Applied and Environmental Science

# Long-Term Temporal Stability of the Resistome in Sewage from Copenhagen

Christian Brinch,[a] Pimlapas Leekitcharoenphon,[a] Ana S. R. Duarte,[a] Christina A. Svendsen,[a] Jacob D. Jensen,[a] Frank M. Aarestrup[a]

[a]Research Group for Genomic Epidemiology, National Food Institute, Technical University of Denmark, Kongens Lyngby, Denmark

**ABSTRACT** Antimicrobial resistance (AMR) is a major threat to global health, and it is crucial to understand the epidemiological aspects in order to predict the emergence and propagation of AMR genes. The aim of this study was to assess the variability and medium-term AMR trends within the mostly healthy human population of a single city. We monitored over 36 months (November 2015 to November 2018) the AMR level in the city of Copenhagen, Denmark, by taking bi-weekly sewage samples from the inlets of the three main water treatment plants, extracting the DNA, performing metagenomic sequencing, and read-mapping against a database of known AMR genes. We found that the AMR level was surprisingly stable with no periodic variability and no signs of drift over the measured period. We found, however, that the seemingly random variations at each site correlate in time with each other, suggesting that the variations we see are due to real environmental changes in the occurrence of AMR.

**IMPORTANCE** The Copenhagen sewage resistome is surprisingly stable in time. The implication is that, at least for cities that are comparable to Copenhagen in terms of sewer infrastructure, few or even single samples provide a robust picture of the resistome within a city.

**KEYWORDS** metagenomics, antimicrobial resistance, microbiome, sewage

Antimicrobial resistance (AMR) is an increasing threat to global health (1, 2), and identifying geographical and temporal trends is crucial to understand the impact on society, determine the effectiveness of control measures, and identify emergence of novel threats needing additional research (1–4).

Traditionally, the surveillance of AMR gene abundances has focused on a few selected pathogens from clinical infections in mainly hospitalized patients based on phenotypic testing (1). However, recently, it has been suggested to utilize human sewage as a way of performing population-wide surveillance of AMR gene abundances in large human populations (5, 6). In the study by Hendriksen et al. (5), metagenomics sequencing of sewage from 79 sites in 60 countries was analyzed, and systematic differences in relative abundance and diversity of AMR genes between Europe/North America/Oceania and Africa/Asia/South America were observed. Furthermore, it was possible to correlate the relative AMR gene abundances found with country-specific socioeconomic, health, and environmental factors, which were then used to perform global prediction on AMR in all countries.

From a routine surveillance point of view, urban sewage is attractive because it provides already anonymized sampling material from a large and mostly healthy population, which otherwise would be both practically, legally, and ethically difficult to monitor. Sewage-based surveillance using metagenomics differs from conventional measures of levels and burden of AMR in several key aspects. It generates pooled data from a large, predominantly nonhospitalized population, whereas the majority of other data refer to hospital patients. The data are also pooled across bacterial taxa in sewage

Address correspondence to Christian Brinch, cbri@food.dtu.dk.

and do not refer to specific, usually cultured, bacterial taxa. Moreover, the resistance genes in sewage do not exclusively originate from human pathogens, or even from human commensals, but also from environmental bacteria, the latter originating mainly from biofilms in the sewer and soil brought by rainwater. Sewage-based AMR data therefore represent a different measure of AMR obtained using a different sampling frame and is, as such, complementary to the current surveillance based on clinical isolates. The surveillance of larger healthy populations and including surveillance of the most common genes might also have some benefits, since it was recently suggested that resistance to front-line drugs is more important than resistance to last-resort antimicrobials (7).

There is so far, to our knowledge, limited understanding of the temporal and seasonal variation of relative AMR gene abundances in human sewage from within single cities and of to what extent single or very few samples from a given country/city may be sufficiently representative for use in global surveillance. Recently, Joseph et al. (8) analyzed wastewater from 14 sites within New York City at three different time points for 7 AMR genes and found that a single time point gave higher relative abundances than the two others. However, it was not possible to determine whether this was natural or true temporal variation.

In this study, we analyzed 312 sewage samples obtained over a time period of 3 years, between November 2015 and November 2018, from the inlets of the three main sewage treatment plants of Copenhagen, Denmark, using metagenomics. The three sites are Rensningsanlæg Avedøre (RA), located at 55°36′30″N, 12°27′01″E, Rensningsanlæg Damhusåen (RD), located at 55°38′26″N, 12°30′20″E, and Rensningsanlæg Lynetten (RL), located at 55°41′42″N, 12°36′53″E. RL covers roughly the urban city center, RD covers the outer boroughs, and RA covers the Copenhagen suburbs. There are some local variations in the population within each catchment area (e.g., variations in ethnicity, income, and demographics), but the population is very homogenous between each of the three catchment areas, with equal access to health care and sanitation. The main difference between the catchment areas is population density, with RL and RD representing a higher population density than RA. Each catchment area serves roughly 300,000 people, so our data sample roughly 1 million people in total, which is approximately 75% of the population in the Copenhagen urban area. The remaining 300,000 people reside in the northern suburbs, which are not covered by either of the three treatment plants, but the population in that region would be comparable to the population in the RA catchment area. The three treatment plants are managed by the same company and operate under the same conditions. There are 2 to 4 hospitals within each catchment area, but they do not discharge untreated sewage. A producer of antibiotics is located in the RL catchment area, only a few kilometers upstream from the inlet. According to their website, they produce vancomycin hydrochloride at their site in Copenhagen. It is not known to us if they treat their wastewater locally before discharging it into the sewer system.

## RESULTS

Visual inspection of the relative abundance of the AMR genes (Fig. 1) showed that the distributions are very similar. Figure 1 provides the distributions of log ratio abundance (Fig. 1a), richness (Fig. 1b), and Shannon index (Fig. 1c) of each site, as well as the Gaussian kernel estimate. A D'Agostino normality test, based on the skew and kurtosis of the distributions, suggests that for both diversity measures, all data are well described by a normal distribution. In the case of the relative abundance, however, the data are not entirely normally distributed, mainly due to the outlying samples with low relative abundance values. Approximately 5% of the samples fall below $3\sigma$, where none or at most a few samples were expected if the relative abundances did follow the best fit normal distribution. Welch's $t$ test suggests that there is no systematic variation in the means of each of these three measures between the three sites, except for a slightly increased relative abundance at the RL site compared to the two other sites, as well as

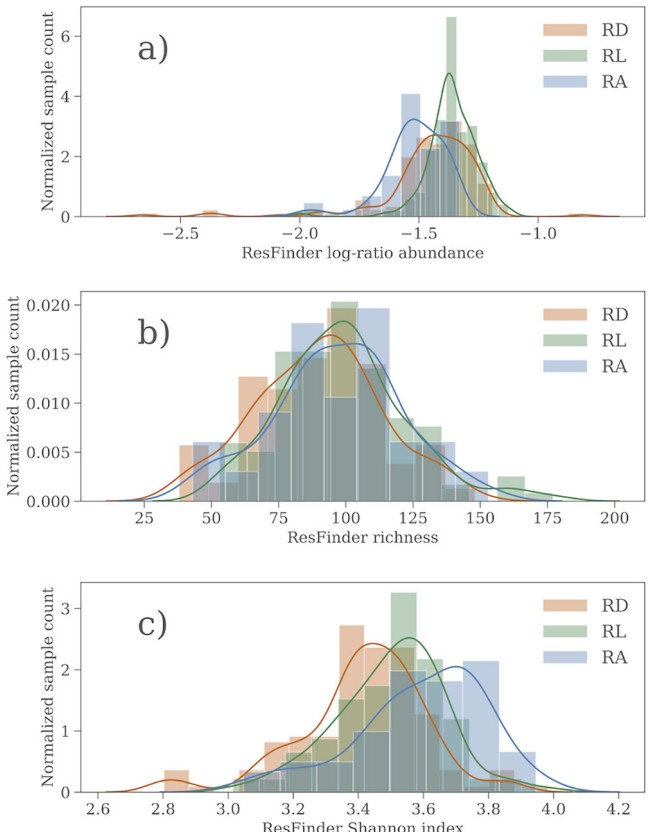

**FIG 1** The sample distribution of derived relative AMR gene abundance and diversity at the three sampling sites. (a) The log-ratio abundance; (b) the AMR gene richness; (c) the AMR gene diversity. The histograms are the estimated values, and the curves are the Gaussian kernel density estimate.

a slightly higher mean in the Shannon index at the RA site ($P < 0.001$ in either case; $H_0$, the mean is the same).

Figure 2 shows the relative AMR abundance in time. In this figure, each sample is shown with its associated uncertainty. Low relative abundance and low sequencing depth contribute to larger uncertainty, so samples that have been sequenced on the MiSeq platform can be identified by the larger error bars (as well as by the colored bars in the top of the plot). There was no systematic variability in time, nor is there any increase or decrease of the mean over the course of 3 years. The data at each of the three sites were consistent with a linear regression slope of zero. The same constant

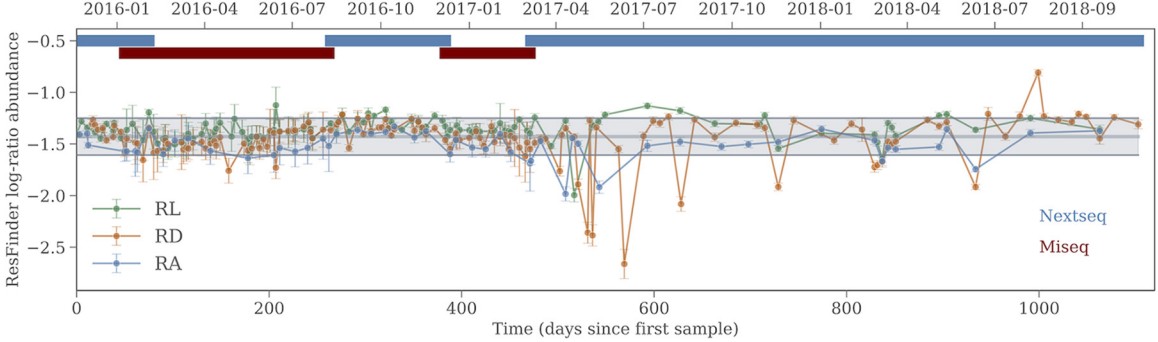

**FIG 2** The AMR log-ratio abundance in time. The points are the transformed counts that result from the read mapping, while the error bars are the estimated counting noise level, which depends on the sequencing depths. The thin gray line striking through the data shows the mean CLR value of the entire data set plus or minus one standard deviation. The colored bars at the top of the plot indicate the sequencing platform which the samples were prepared on.

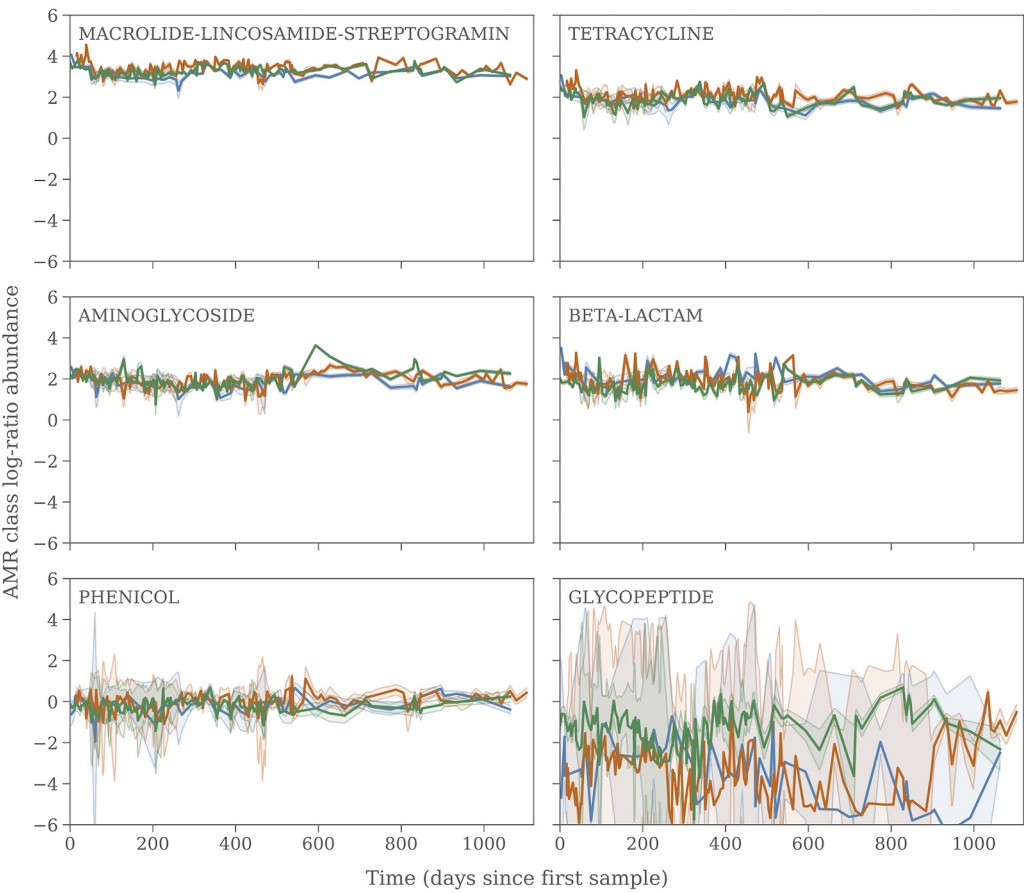

**FIG 3** Similar to Fig. 2 but for counts aggregated into antimicrobial resistance (AMR) classes.

trend was seen in the time-resolved diversity measures. However, what appears to be random day-to-day variation did, in fact, have a (weak) positive correlation between each pair of sites. The average Pearson correlation coefficient for the relative abundance between sites and for the AMR richness between sites was 0.3 for both measures, whereas for the Shannon index between sites, the correlation coefficient was 0.5. The implication is that the variation seen in Fig. 2 is not due to noise but is, in fact, due to real AMR changes in the city. It should be pointed out, though, that the correlation presented here is only calculated for the samples that have been taken on the same day at each pair of sites. These samples make up about 70% of all of the data, so in reality, the correlation coefficient could be different. The strongest and most significant correlation for the relative abundance is between sites RL and RD, where we have 90 samples in the intersection of sample days, which corresponds to about one third of the entire data set. The correlation coefficient is 0.27 with a $P$ value of 0.009 for the null hypothesis that the two sets of samples are drawn from uncorrelated distributions.

If we aggregate the ResFinder counts to the antibiotic class, which the genes give resistance to, we can track the proportion of the resistome that is made up of the various resistance classes. Figure 3 shows the 8 most abundant resistance classes. We furthermore identify AMR genes that give resistance to 10 additional antibiotic classes, phenicol, streptogramin A, pleuromutilin, quinolone, fosfomycin, rifampin, fluoroquinolone, polymyxin, oxazolidinone, and folate pathway antagonists. These, however, have very low relative abundances and are in many cases dominated by false-positive read mappings, Bayesian zero-replacement noise, and low number statistics, so even if some of these genes are actually present in the environment, we do not consider them in the subsequent analysis. Two things are immediately apparent: (i) each part of the composition is surprisingly constant in time, and (ii) there is virtually no variation between

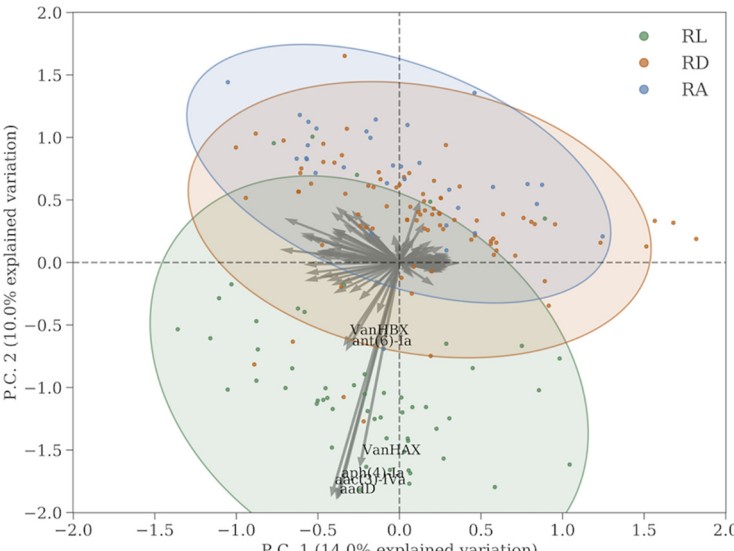

**FIG 4** Ordination plot showing the projection of the AMR gene CLR values onto the plane spanned by the two principal components (P.C.). The colors correspond to sample sites, and points represent a similar projection of the samples. The ellipses describe the 90% confidence region. Only the main features have been labeled.

sites, at least not above our current detection limit. Only glycopeptide shows a clear excess at the RL site with respect to the two other sites, on average by a factor of almost 10. The composition of the resistome is, by and large, constant in time.

We investigated the variance in individual resistance genes further by plotting the first two principal components of the subcomposition from Fig. 3 in a biplot, shown in Fig. 4. From the biplot, we see that by far the largest contributor to variance in our samples are the VanHAX/HBX genes, which give resistance to glycopeptide, as well as a handful of genes giving resistance to aminoglycoside. The latter genes are not enough to show a significant overabundance of aminoglycoside resistance at RL compared to the other two sites. The samples are shown as colored dots in the biplot. The 90% confidence region is shown as colored ellipses. It is clear that there is no significant difference in the distribution of points from RD and RA, whereas the samples from RL are clearly associated with a few genes.

Figure 5 shows plots of the log-ratio abundance versus the mean relative abundances of the resistance genes (MA plot) at each of the three sites versus the two other sites. The panels show the genes (marked by red dots) that are significantly differentially abundant (using a two-sided Welch's $t$ test with a confidence interval of $1 - \alpha = 99.7\%$) with respect to the other two sites, and these are also labeled with their names and their phenotypic resistance class. We identified four different genes that are overabundant at RL, four that are significantly underabundant at RD, and a few over- and underabundant genes at RA. Apart from the glycopeptide resistance gene, which gives rise to the increased glycopeptide resistance in RL seen in the previous figures, we find a number of aminoglycoside resistance genes that are differentially abundant at RL. Their median abundances, however, are low, so they do not contribute considerably to the overall aminoglycoside resistance pressure.

Finally, we looked at the Copenhagen sewer bacteriome. More than 1,500 different genera of bacteria were detected, many of which, however, at a very low relative abundance. We found for the bacteriome that, like the resistome, it was very stable across the city and in time as well. The 30 most abundant genera account for about 80% of all bacterial reads, and these are listed in Table 1. Only 1 genus showed systematic variance across the city, namely, *Streptomyces*, which is strongly associated with the RL site. Figure 6 shows the biplot of a principal-component analysis of the 30 most abundant bacterial genera, in which the association between *Streptomyces* and

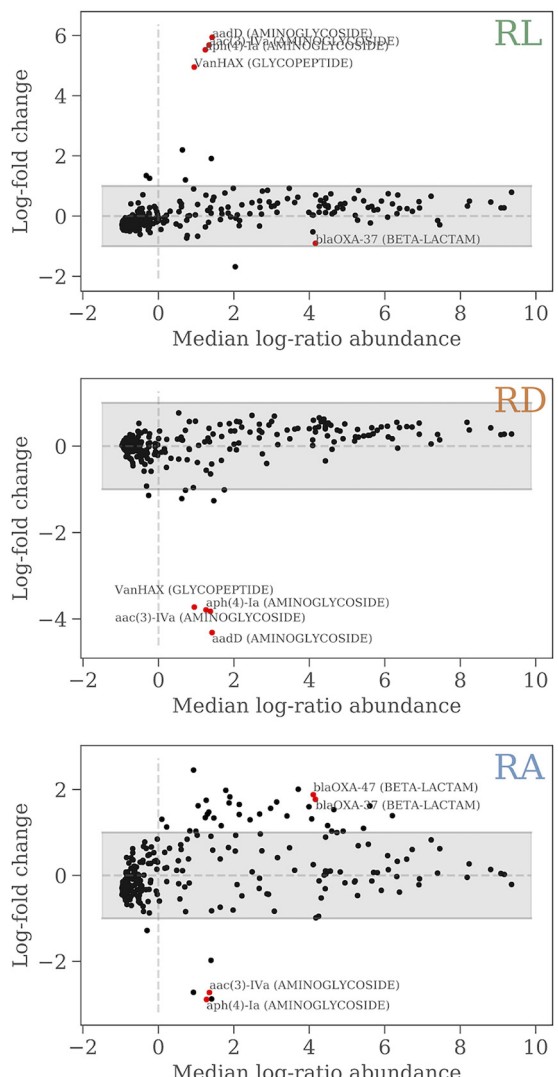

**FIG 5** Plots of the log-ratio abundance versus the mean relative abundance of resistance genes at each site versus the two other sites. Genes are represented by points, and the red points are the (named) genes which are differentially abundant. The gray shaded area covers a single log-fold change.

the RL samples is clear. Species of *Streptomyces* are known antibiotic-producing organisms, and therefore the excess of glycopeptide resistance genes as well as the excess of *Streptomyces* could be explained by the fact that an antibiotic-producing factory is located within the catchment area a few kilometers upstream from the water treatment plant (9, 10).

## DISCUSSION

The limited variation between our samples proves the strong homogeneity of the resistome within the city of Copenhagen, both over time as well as across the city. This may well reflect the fact that Copenhagen has a very homogeneous population in terms of demography and living standards and also that Copenhagen has a very uniform water treatment infrastructure. Despite the fact that antibiotic usage (AMU) does vary over the course of a year, with higher consumption during winter and early spring (11), we do not see any annual variation in the relative AMR gene abundance. Joseph et al. (8) did observe a higher relative abundance in five of seven genes tested for in the month of May compared to February and August and suggested that this might be due to a delayed effect of higher AMU in February and March. However, since

**TABLE 1** Top 30 highest relatively abundant bacterial genera

| Genus name | Mean CLR | SD | Minimum CLR | Maximum CLR |
| --- | --- | --- | --- | --- |
| Pseudomonas | 8.80 | 1.60 | 5.01 | 14.02 |
| Acinetobacter | 10.85 | 0.96 | 7.31 | 13.73 |
| Psychrobacter | 7.29 | 1.62 | 2.39 | 13.27 |
| Shewanella | 3.71 | 2.73 | −1.22 | 12.67 |
| Bacteroides | 10.11 | 0.96 | 7.15 | 12.64 |
| Janthinobacterium | 0.13 | 3.84 | −3.37 | 12.53 |
| Streptomyces | 2.13 | 3.47 | −1.59 | 12.44 |
| Campylobacter | 9.07 | 1.01 | 6.25 | 12.38 |
| Acidovorax | 10.87 | 0.60 | 7.91 | 12.26 |
| Faecalibacterium | 9.50 | 0.98 | 4.13 | 12.13 |
| Arcobacter | 8.69 | 1.00 | 5.32 | 12.12 |
| Aeromonas | 8.35 | 1.39 | 3.76 | 12.09 |
| Prevotella | 8.84 | 0.90 | 5.05 | 11.78 |
| Lactococcus | 9.35 | 0.99 | 4.60 | 11.38 |
| Sulfurospirillum | −0.04 | 2.49 | −3.19 | 11.35 |
| Parabacteroides | 8.63 | 0.80 | 6.39 | 11.32 |
| Eubacterium | 8.85 | 0.88 | 5.11 | 11.22 |
| Escherichia | 8.05 | 1.14 | 4.57 | 11.13 |
| Moraxella | 8.01 | 1.04 | 3.90 | 11.00 |
| Enterobacter | 6.45 | 1.74 | 2.29 | 10.99 |
| Enterococcus | 8.19 | 0.93 | 4.67 | 10.94 |
| Alistipes | 8.61 | 0.82 | 5.70 | 10.88 |
| Moraxellaceae | 7.21 | 1.41 | −1.44 | 10.88 |
| Citrobacter | 7.10 | 1.63 | 0.92 | 10.71 |
| Trichococcus | 9.21 | 0.80 | 6.75 | 10.71 |
| Clostridium | 8.99 | 0.81 | 6.06 | 10.66 |
| Giesbergeria | 4.38 | 1.09 | 2.20 | 10.60 |
| Streptococcus | 8.16 | 1.46 | 2.70 | 10.54 |
| Cloacibacterium | 7.76 | 1.16 | 4.16 | 10.49 |
| Dialister | 8.71 | 0.89 | 4.65 | 10.46 |

they only compared three time points, it is difficult to determine whether their observed differences are simply natural random variation.

Other potential sources of variation are the increased number of tourists during summer and the possible annual change in the microbiome due to the seasonal change in ambient temperatures. We see no evidence for either effect in the resistome. Changing diet could also lead to variations in the resistome, but again, such changes are either cyclic with a 1-year period, because of seasonal availability of certain food items, or due to long-term population trends such as increasing veganism or just decreasing meat consumption. We observed neither periodic variability nor linear trends, so either these effects are not affecting the resistome, or the influence happens below our detection limit.

Precipitation has been suggested as a source of variation in the AMR footprint by changing the composition of the sewage, but historic data from the European Climate Assessment & Dataset (ECA&D) (12) of the daily rainfall in Copenhagen shows no correlation with the relative AMR abundance. Figure 7 shows the relative total abundance of AMR against rainfall in Copenhagen. Because AMR levels are not expected to change instantaneously and because sewage can have up to 12 hours of travel time before it reaches the treatment plants, we sum the rainfall over the last 3 days before the sample day. We see no hints of association between rainfall and relative AMR abundance. However, our comparison is simplistic because we do not take the location of the rainfall within each catchment area, the concentration of the rain (light drizzle over the entire day versus short bursts of heavy rain), or the time of day, which affects the flow rate in the sewage system, into account, which would require advanced modeling and is beyond the scope of this paper.

Still, we find that the variation within each site correlates with the others to some extent, which means that the fluctuations that we see are not only due to sample collection or processing noise, but at least in part, are due to actual, city-wide variations in the environment. Although there could still be some randomness involved, it is more

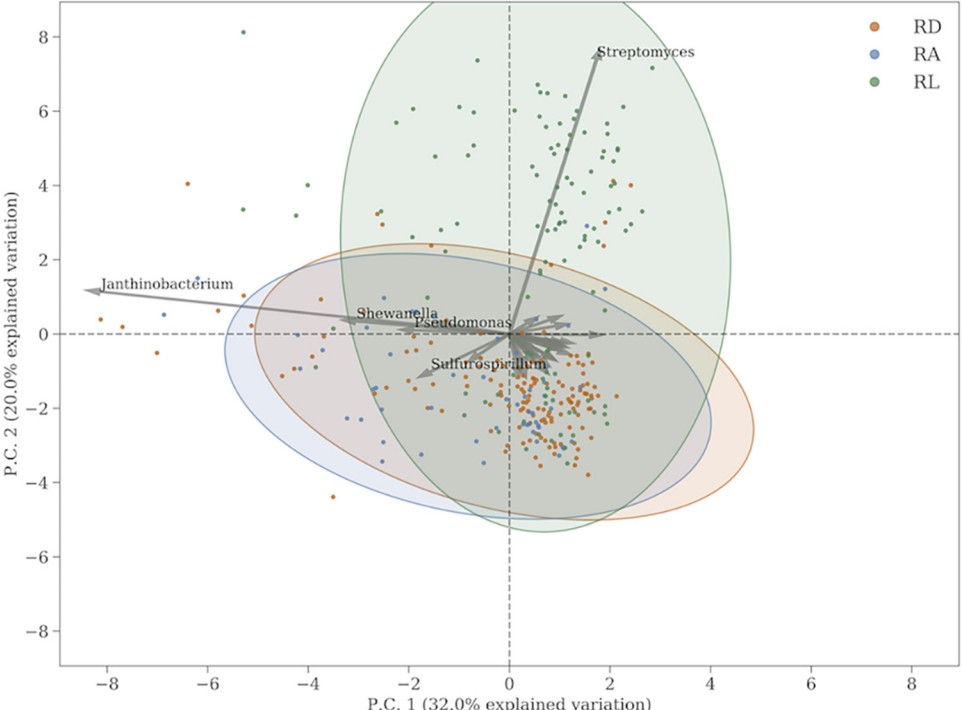

**FIG 6** Similar to Fig. 4 but for bacterial genera.

likely that the AMR levels are causally governed by a set of variables that are still to be identified.

The resistome is largely consistent across the sampling sites, but critical differences exist among a few low-abundance but clinically relevant AMR genes, primarily the VanHAX gene cassette. VanHAX gives resistance to vancomycin, which is a type of glycopeptide antibiotic, and these genes drive the differences found in glycopeptide resistance across Copenhagen. Apart from this, our data show that the AMR level in Copenhagen can be adequately described by a single mean value and a standard deviation (see Fig. 1). We have also previously observed that using a limited number of samples from the same sites as well as multiple samples within a city is more similar than using samples obtained from different sites and cities (5). It would be interesting

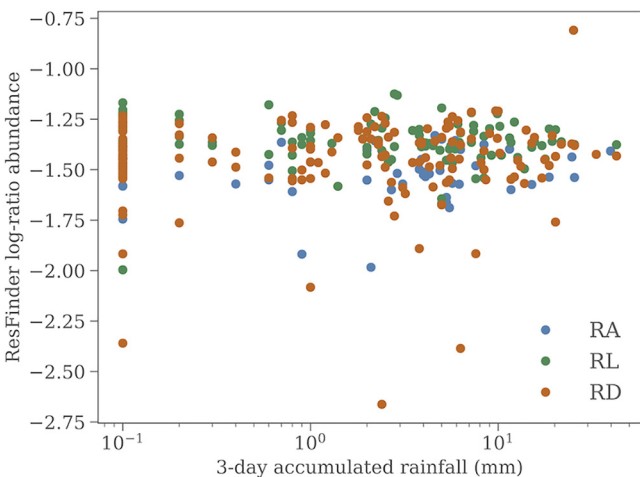

**FIG 7** The relative antimicrobial resistance gene abundance at each of the three sites as a function of accumulated rainfall over 3 days before the sample day.

to see whether further studies confirm these findings, which would imply that we can quantify the AMR variation between cities using few samples.

## MATERIALS AND METHODS

**Samples.** Samples were collected at nonregular intervals at a rate of 1 to 2 per week per site. Each sample was drawn over the course of 12 hours by slowly filling a bottle, drop by drop, from the main flow pipe of the inlet. A sample therefore represents the average sewage composition over half a day. All samples were frozen at the treatment plant and kept at subzero temperatures until processing in the laboratory. All three plants used a similar sampling technique, and all samples were acquired before any mechanical, biological, or chemical treatment of the sewage.

**DNA purification and sequencing.** As of 1 June 2019, 312 samples had been processed in the laboratory, with 54 samples coming from RA, 144 samples coming from RD, and 114 samples coming from RL. This amounts to approximately 50% of all samples collected. Up until April 2017, all collected samples got sequenced, whereas after this date, samples were selected for sequencing in order to obtain a uniform time sampling and site coverage and maximize the sample period and the sequencing depth within our sequencing budget. After defrosting, the water was centrifuged and the sewage was collected as pellets, from which DNA was extracted. All libraries for sequencing were made using the Illumina Nextera XT DNA library preparation kit following the manufacturer's instructions. All sequencing was done in-house, with 138 samples sequenced on the MiSeq platform and 174 sequenced on the NextSeq platform. The change of sequencing platform was circumstantial and was not part of the experiment design. Samples were processed in blocks on each platform, and it is indicated in Fig. 2 which platform sequenced the samples. On average, the NextSeq samples were sequenced to a depth of $2.3 \times 10^7$ reads, while the MiSeq samples were sequenced to a depth of $3.2 \times 10^6$ reads, a difference of 1 order of magnitude. This poses a problem for the diversity measures when comparing samples across sequencing platforms, since the more deeply sequenced NextSeq samples find many more low-abundance genes and thus have a much greater richness. Subsequent analysis of the read mapping results takes varying sequencing depth into account by treating the data compositionally, but certain properties, such as richness and diversity, are not directly comparable between samples from the two platforms. However, the proportions of samples sequenced on the two platforms are the same for each site, so they are comparable between sites.

**Bioinformatics and read mapping.** After DNA sequencing, the reads were trimmed, including adaptor removal, using BBduk2 36.49 (13) with the quality threshold set at 20 and a minimum length of 50 bp. Trimmed reads were used as input to the reference-based mapping and taxonomy-assignment tool KMA 1.2.10a (14), which uses global alignment to assign the reads to a reference database. We used a preset alignment score over an alignment length threshold of 0.5, resulting in an effective identity of around 90%. An acquired AMR gene database (ResFinder [15], downloaded January 2020) was used to annotate reads. The AMR genes were of bacterial origin and could therefore align to both bacterium databases and the ResFinder database. The bacterial content of the samples was determined by mapping the reads to the 16S Silva database (https://www.arb-silva.de, downloaded January 2020) and summing the hits to bacterial references. In addition, we mapped the reads to a range of bacterial reference databases, bacteria (closed and draft genomes, NCBI GenBank, April 2019), MetaHitAssembly (PRJEB674 to PRJEB1046, April 2019), and HumanMicrobiome (genome assemblies, NCBI GenBank, April 2019) in order to characterize the bacteriome. On average, the MiSeq samples had $1.6 \times 10^4$ fragments mapped to 16S, $2.7 \times 10^5$ to the genomic databases, and 884 fragments mapped to ResFinder of a total of $2.6 \times 10^6$ fragments sequenced. The average values for the NextSeq samples were $1.3 \times 10^5$ fragments mapped to 16S, $3.3 \times 10^6$ to the genomic databases, and $7.8 \times 10^3$ to ResFinder out of a total of $2.5 \times 10^7$ fragments.

**Data processing.** All subsequent data analysis has been done using Python3 and Pandas/SciPy. Due to the compositional nature of the mapped data, we applied appropriate log-ratio transformations (16) when needed. We use the centered log-ratio (CLR) transform, defined for a *d-part* composition **x** as

$$\mathrm{clr}(\mathbf{x}) = \left[ \frac{x_1}{g_m(\mathbf{x})}, \frac{x_2}{g_m(\mathbf{x})}, \dots, \frac{x_D}{g_m(\mathbf{x})} \right], \quad g_m(\mathbf{x}) = \left( \prod_{i=1}^{D} x_i \right)^{1/D}$$

The composition consists of a part for each ResFinder reference, amalgamated to resistance class level, and a part for the sum of 16S counts. For the analysis involving the overall relative AMR abundance, we amalgamate all the ResFinder counts to form a two-part composition with the 16S counts. Before transforming the counts, we adjust for differences in gene length by dividing the counts of individual genes by the length of the gene, thus correcting for the effect that long genes produce more reads than shorter genes of equal relative abundance. After this correction, each gene count carries the same weight. We also homology-reduced the ResFinder genes by summing counts to genes that are more than 90% identical. The ResFinder count matrix is very sparse, with 75% of the entries equal to 0. The sparsity is a serious problem for the log-ratio transformations, because the logarithm of zero is undefined. We chose to use Bayesian inference to replace the zeros, where we make the assumption that, during sequencing, reads are drawn randomly from the DNA pool, so that the resulting proportions of counts follows a multinomial distribution. We then used the observed count vectors as weights for a uniform Dirichlet distribution, the conjugate prior for the multinomial distribution, from which we then drew several thousands of random instances to form the posterior distribution of probabilities for the count number of each gene. Not only do we get a nonzero estimate for the count numbers, even when zero counts were observed, but we also get a count uncertainty associated with each gene, with the uncertainty going down as the total number of observed count increase. This Bayesian method relies on

the implicit assumption that all zeroes are rounded zeroes, and no zeroes are essential. This is likely not always true. In fact, the more deeply a sample has been sequenced, the more likely it is that any zero entry is an essential zero and not a rounded zero, and even our deepest sequenced samples contain many zero entries. We therefore disregard the resistance classes in our analysis, which consists of genes that are predominantly nondetected, due to the large uncertainty in their relative abundance.

**Data availability.** All raw sequence data have been uploaded to the European Nucleotide Archive (ENA) under accession number PRJEB34633. Accession numbers and metadata for the individual samples can be found in the supplemental material (Table S1).

## SUPPLEMENTAL MATERIAL

Supplemental material is available online only.

**TABLE S1**, XLSX file, 0.04 MB.

## ACKNOWLEDGMENTS

We thank the staff at BIOFOS for providing access to the sewage treatment plants and for collecting the samples on a daily basis.

This study has received funding from the Novo Nordisk Foundation (NNF16OC0021856: Global Surveillance of Antimicrobial Resistance). F.M.A. conceived the experiment and secured the funding, J.D.J. collected samples, C.A.S. did sample preparation and sequencing, P.L. and C.B. did the bioinformatics, C.B. and A.S.R.D. analyzed the data, and C.B. and F.M.A. wrote the paper. All authors reviewed the manuscript.

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
