## [Reviewer comments · mSystems]

Long-term temporal stability of the resistome in sewage from Copenhagen

Christian Brinch, Pimlapas Leekitcharoenphon, Ana Duarte, Christina Svendsen, Jacob Jensen, and Frank Aarestrup

Corresponding Author(s): Christian Brinch, Technical University of Denmark

Review Timeline:

Submission Date:

August 21, 2020

Accepted:

September 29, 2020

Editor: Anthony Fodor

Reviewer(s): The reviewers have opted to remain anonymous.

Transaction Report:

DOI: <https://doi.org/10.1128/mSystems.00841-20>

Response to reviewers

Changes in the manuscript are highlighted. Changes related to comments by reviewer 1 in blue and by reviewer 2 in red. Comments which apply to both reviewers are marked in yellow. Two major concerns were raised by both reviewers: (1) that we used a subset of the data to determine correlation between sites, and (2) the unjustified mentioning of *Streptomyces*.

(1) Correlating time series which are sampled on uneven and unequal time intervals is notoriously difficult, particularly when the signal is noisy or non-periodic. There is simply no way in which we can infer the values on days where no samples were taken. Interpolation is meaningless because variation occurs on the smallest time scales (1 day) and there is no guarantee that the interpolated value reflects the real (unknown) value. Binning the data on week intervals does not help, since it averages a lot of the fluctuations out and quite often, we are missing a whole week of samples from a site and therefore this kind of binning does not add many more points to correlate. In fact, it lowers the number of data points, because many of the common sample days are averaged together. It may be possible to predict missing values using autoregression methods, such as ARIMA or maybe Kalman filtering, but these rely on a working model for the behavior, or at least periodicity, of the signal and we have neither. The editor suggests using autocorrelation. It is my understanding that autocorrelation is used to find repeating patterns within a single time series. Could it be that the editor is thinking of cross-correlation? Cross-correlation may be an option, since cross-correlation looks for similarity between two different time series as a function of displacement between the two. However, I have not been able to find implementations of cross-correlation methods for uneven sampled data which does not rely on interpolation, which brings us back to the problem of variability on the smallest scales.

That said, I do not think that our results, obtained by correlating the signal on the sample day intersections, is as bad as the wording in the manuscript suggests. I just realized that the text says “a relatively small subset” which, given the numbers I quote below, is rather inaccurate. I never actually considered what fraction of the samples went into this calculation. I have changed the wording and added the actual numbers in the text.

Let me elaborate a little bit on the statistics here:

We have 54 samples from RA, 114 samples from RL and 144 samples from RD. The sample day intersection between RL and RD results in 90 days on which samples were taken in both places. That corresponds to 79% of all samples from RL and 64% of all samples from RD. For the cases RA vs RD and RA vs RL, the numbers are 37 and 36 common sample days respectively corresponding to 69% and 67% of the samples from RA and 26% and 32% of the samples from RD and RL. The latter low percentages are simply due to the fact that there are many more samples taken from RL and RD than from RA. To summarize, around 70% of all of our samples are utilized for this exercise. This is not “a relatively small subset”, so that was really a poor choice of words from my side. It is correct that the correlation is rather weak, and we do not state otherwise in the manuscript. Pearson R is around 0.3 (was 0.4 in the manuscript, but I corrected a minor mistake) for all three combinations of sites. The fact that all three comes out with the same correlation coefficient gives the result some merit: had it been +0.3, 0, and -0.3 I would be much more reluctant about making the claim that the sites are (weakly) correlated. We can also calculate p-values for the zero hypothesis that the samples are drawn from distributions with zero correlation. For RA-RD and RA-RL those

p-values are 0.047 and 0.129, that is, the correlation is borderline and not significant. This, however, is largely due to the low number of points (37 and 36) for these two cases. For RD-RL, where we have 90 common sample days, the p-value is 0.009. The fact that we have two significant correlations with the same coefficient adds to the robustness of the result. I do not think there is much more we can do with the data at hand without introducing a lot of assumptions. We have provided the additional information to the manuscript about the number of points used in these calculations. We hope that this resolves the concerns of the reviewers and the editor.

(2) An earlier version of this manuscript was submitted to a different journal and went through review after which we retracted the manuscript due to disagreements with one of the reviewers (he/she disagreed with our compositional approach and we refused to treat the data in a non-compositional way). We originally had mapped all reads against a bacterial database and used the total bacterial count for normalization. We also had some treatment of the bacteriome in that version. However, that previous reviewer wanted us to use 16S as a proxy for bacteria and remove the part involving the bacteriome. It is my mistake that the sentence mentioning *Streptomyces* was left in the manuscript without justification. We have remapped all our data against a bacterial genomic database and re-included a (brief) treatment of the bacteriome and this explains our knowledge of the *Streptomyces* abundance.

Please find below a point by point response to each reviewer.

Reply to reviewer #1

Major comments

1) See response above about correlation.

2) Line 70 -87: We have added a paragraph (in yellow) to the introduction, providing more detail. We give rough numbers, because the catchment areas are not perfectly well-defined. Flow can be redirected in the sewers during maintenance or close to capacity limits. Also, the catchment areas do not follow administrative boundaries exactly, so it is difficult to give exact population figures.

3) There is an ongoing effort to separate rainwater from sewage throughout the entire country of Denmark. The main reason for this is to preserve sub-surface drinking water reservoirs (all drinking water is pumped from the underground in Denmark) by letting rainwater sift through the soil layer rather than carry it through the sewers to the WWTP which releases it (after treatment) into the sea. However, the WWTPs do see significant volume changes at the inlet during rainfall, but it is hard to quantify the fraction of the wastewater which is rainwater, because it is masked by other effects such as day/night changes in toilet usage, flow rates being regulated by pumping stations within the catchment area, etc. Do we actually state that we are looking for temporal changes in AMR levels in the environment? I can't find any mentioning of this and this is certainly not our intention. The reason why we look at correlation between rainfall and AMR is to see if the effect of diluting sewage or flushing the sewer with rainwater has an effect on the signal. We have made that more clear in the text.

4) Line 159-171: We do not claim that *streptomyces* carry the vanHAX/HBX gene cassette. However, the antibiotics producer in the catchment area states on their website that they

use streptomyces to produce vancomycin. VanHAX gives resistance to vancomycin, and we happen to find those genes as well as an excess level of Streptomyces downstream from the plant. That could be pure coincidence or it could be related. We don't have the data to tell, but it is a possible that there is a connection. *If* the excess streptomyces originates from that plant, I do not want to speculate in the manuscript how it got into the sewer (but I can certainly think of ways). The vanHAX/HBX genes could have a completely different route into the sewer. For instance, plant workers could have prolonged exposure to vancomycin and their microbiomes could have accumulated resistance towards it, which would then be shed into the toilets at the factory. This is speculation, and I don't want to point fingers at anyone in a scientific publication. Our point (which I have tried to make more clear in the manuscript) is not to report or identify a specific source of the AMR genes, but rather that by doing metagenomics on urban sewage, it is actually possible to see explainable effects. It is not (all) just random noise which makes everything a grey mess. When a (possible) source exists, we actually see an effect in the sewage. I hope that by including the treatment of the bacteriome, we have resolved the reviewers concern on this point.

Minor comments

- 1) Line 45: We agree. We have softened the phrasing.
- 2) Line 49-50: These are mainly associated with biofilms in the sewer but also soil which is washed off with rain, and some freshwater bacteria which probably originates in open rainwater reservoirs, that are sometimes used to "flush" the sewer. A comment has been added.
- 3) We agree. We have simply cropped the sentence.
- 4) Line 60: The text has been changed according to the suggestion by the reviewer.
- 5) Line 70-87: We have expanded the description of the WWTP in the introduction (in yellow).
- 6) The sentence has been deleted from the introduction.
- 7) Line 89: Yes. Reads that are assigned to a resistance gene reference are by definition bacterial. We have changed the wording to avoid confusion.
- 8) Line 99: The text has been changed according to the suggestion by the reviewer.
- 9) "drift" is defined in mathematics as the linear term in a stochastic process, i.e., the change in the mean between any increment. We agree the term "substantial" is vague, so this word has been removed. We have kept "drift" in the abstract but changed the wording to "increase or decrease of the mean" within the text.
- 10) Line 126-129: We have rephrased the sentence.
- 11) Line 133: "parts" refer to the composition. We have emphasized that.
- 12) Line 138-139: Changed the wording.
- 13) Line 147-148: Defined MA
- 14) Line 190: The text has been changed according to the suggestion by the reviewer.
- 15) Line 194-205: We rearranged the sentences as suggested.
- 16) We have removed this part due to comments from the other reviewer.
- 17) This is speculation. We have removed the sentence.
- 18) Line 253-254: Yes, the proportions are the same. We have noted this in the text.
- 19) Fig 1: Probably a poor choice of words. It is just a normalized histogram. I have changed the label to normalized number of samples.
- 20) Fig 3-5: Done.

21) supplement: To be honest, I have no idea. That sheet was provided by our lab technician and I have overlooked it. I have removed the blue shading. There is nothing special about these samples.

Reply to reviewer #2

1) Line 92-93: I used Scipy's normaltest which is based on D'Agostino and Pearsons omnibus test for normality. A K-S test and even a Gauss fit and a χ^2 test gives the same result. I have updated the text.

2) We agree, relative abundance is, of course, the proper term. Overall AMR abundance is the CLR transformed proportion of total ResFinder and total 16S counts. We have clarified this in the text in the section "Data Processing" and abundance has been changed to relative abundance throughout the text (when appropriate).

3) Line 96-98: This is based on z-scores and the expected number of samples within increasing confidence intervals. Given the number of samples, we would not expect to find any samples below 3sigma, but 5% or so of the samples do fall below 3sigma. We say approximately 5% because the standard deviation is calculated from all sample (assuming normality and thus including these outliers) and is therefore potentially inflated. It is not important for the further analysis whether it is 4% or 6% of the samples which do not follow a normal distribution and we have therefore not bothered with anything more advanced, such as outlier detection or such. We have changed the wording a bit to clarify the approach.

4) Line 112-115: Correct. We have made that clear.

5) These are the same subsets as discussed above

6) Genes that give resistance to multiple classes are counted in each category. It is correct that the majority of genes between the three mentioned classes are the same, but not all of them. Under other circumstances, we would argue that presenting these classes individually is more valuable, but we agree that given the lack of anything interesting happening in our data in either of the three classes, we can save some real estate by consolidating them. We have merged the three classes, by counting all macrolide reads and only the Lincosamide and streptogramin counts that are not in the macrolide class already, so that nothing is counted twice. This removes two panels from Fig 3.

7) False positives read mappings are not frequent, but it can happen. We do not explicitly quantify the effect, because even if we could remove them, we would still not trust small counts. An example of false read mapping was a sample (from another study) with a very high *Yersinia Enterocolitica* count (tens of thousands) and 3 reads that mapped to *Yersinia Pestis*. It was very unlikely that the sample (a food item from a Danish supermarket) actually contained *Y. Pestis*, but the three reads did in fact map better to the *Pestis* genome than to the *Enterocolitica* genome, however, only by a few bases. Whether these were sequencing errors, point mutations, errors in the reference or something else is not known, but an outbreak of plague was not reported. The 3 read were simply discarded.

8) See the general comment above. We have remapped all our data against a bacterial genomic database and re-included a (brief) treatment of the bacteriome and this explains our knowledge of the *Streptomyces* abundance. We continue to use 16S for normalization,

given that 16S correlates almost perfectly with the bacteria count and we would have to redo everything in the paper to go back to using bacteria, with no difference to the conclusions. There are small additions here and there in the manuscript, as well as a larger section highlighted in yellow, relating to the inclusion of the bacteria mapping.

9) We have added some text to the introduction, where we also mention the antibiotics producer (in yellow).

10) We have replaced the class PCA with a gene PCA as requested. The text has been updated to reflect this.

11) 99.7% is 3sigma. That is a choice. We happen to find 366 different ResFinder references. That gives an error rate of ~ 1 , when testing on a 3sigma confidence level.

12) Line 212-216: Certainly. We have adopted the formulation provided by the reviewer. We detect the entire VanHAX cassette. For reasons unknown to me, the entire cassette is given as a single reference in ResFinder, but a quick inspection shows that we get an even coverage and low variation across the entire reference, i.e., all three genes are present and co-abundant.

13) I am not entirely sure I agree with the reviewer on this point, but since the other reviewer had issues with this analysis as well, we have left it out.

14) Fig 6 & 7: We agree Figures 6 and 7 has been merged and the text has been adapted accordingly.

15) Line 244-246: Sure. This information has been added.

16) Line 274-278: The function (a sum) over i is not missing. By convention, summation over repeated indices are implied (Einstein notation). However, there is no need for fancy notation here I realize, so I have rewritten the formula as it is usually presented.

September 29, 2020

Dr. Christian Brinch
Technical University of Denmark
National Food Institute
Kgs. Lyngby
Denmark

Re: mSystems00841-20 (Long-term temporal stability of the resistome in sewage from Copenhagen)

Dear Dr. Christian Brinch:

Your manuscript has been accepted, and I am forwarding it to the ASM Journals Department for publication. For your reference, ASM Journals' address is given below. Before it can be scheduled for publication, your manuscript will be checked by the mSystems senior production editor, Ellie Ghatineh, to make sure that all elements meet the technical requirements for publication. She will contact you if anything needs to be revised before copyediting and production can begin. Otherwise, you will be notified when your proofs are ready to be viewed.

While your paper is acceptable in its current form, I have attached for your reference comments from one of the reviewers. As you edit the final manuscript for publication, you may wish to consider their comments. I also here note a very minor typo in line 200 where "Because AMR levels are not expected to change instantaneous" should probably be "Because AMR levels are not expected to change instantaneously"

Sincerely,

Anthony Fodor
Editor, mSystems

Comments Reviewer #3:

Many thanks to the authors for updating the manuscript based on the previous round of feedback. These have provided greater clarity on the analyses conducted and their interpretation. In particular, the further description of the same day sampling pairs and the different sites was appreciated. For the most part, my concerns have been resolved.

I appreciate the additional consideration of the bacterial composition now included in the manuscript, albeit very briefly. I was hoping for an exploration of any associations between taxa and ARGs/ARG classes, as suggested previously. In particular, is the vancomycin resistance correlated with *Streptomyces* relative abundance?

Line 121 - perhaps include the % of total samples that 90 represents here in the text.

I like the additional language included in the conclusion (line 212) regarding the importance of gene-level differences. If possible, it would be good to modify the importance statement slightly in a similar fashion. I think you could get a pretty robust characterization of the sewage resistome in Copenhagen from a few samples at each site, but considering only one site could miss high abundances of clinically relevant ARGs, such as the van cassette.